# Effect of Iron and Folic Acid Supplementation on the Level of Essential and Toxic Elements in Young Women

**DOI:** 10.3390/ijerph18031360

**Published:** 2021-02-02

**Authors:** Joanna Suliburska, Agata Chmurzynska, Rafal Kocylowski, Katarzyna Skrypnik, Anna Radziejewska, Danuta Baralkiewicz

**Affiliations:** 1Department of Human Nutrition and Dietetics, Poznan University of Life Science, ul. Wojska Polskiego 31, 60-624 Poznań, Poland; agata.chmurzynska@up.poznan.pl (A.C.); katarzyna.skrypnik@up.poznan.pl (K.S.); anna.radziejewska@up.poznan.pl (A.R.); 2PreMediCare New Med Medical Center, ul. Drużbickiego 13, 61-693 Poznan, Poland; rkocylow@gmail.com; 3Department of Trace Element Analysis by Spectroscopy Method, Faculty of Chemistry, Adam Mickiewicz University in Poznan, ul. Umultowska 89b, 61-614 Poznan, Poland; danutaba@amu.edu.pl

**Keywords:** iron, folic acid, supplementation, elements, women

## Abstract

Although simultaneous supplementation of iron and folic acid is justified, the potential interactions between these micronutrients and other elements are poorly known. In this study, we aimed to investigate the effect of iron and folic acid supplementation on the levels of selected essential and toxic elements in the serum of micronutrient-deficient young women. A total of 40 women participated in this study and were divided into two groups: study group (*n* = 23) (with iron and folate deficiency) and control group (*n* = 17). The study group received iron and folic acid supplements for 3 months. Blood samples were collected at baseline and after the completion of the study period. Women completed a 3-day food intake record. We calculated the body mass index (BMI) of all the participants. Cellular morphology was analyzed in whole blood, and biochemical parameters were determined in serum. Elements were measured in serum by inductively coupled plasma mass spectrometry (ICP-MS). According to our results, in the case of the study group, the supplementation of iron and folic acid restored their levels; however, it caused a significant decrease in the level of zinc, calcium, and magnesium. In the case of the control group, at the end of the study period, there was a marked decrease in the level of iron. Interestingly, there was an increase in the level of arsenic and vanadium in both groups. In conclusion, simultaneous supplementation of iron and folic acid impairs the level of zinc, calcium, and magnesium in women of childbearing age.

## 1. Introduction

Iron and folic acid deficiency are common in young with ages ranging from 15 to 49 years of age, and pregnant women [1]. Therefore, the World Health Organization (WHO) has recommended a daily oral supplementation of iron and folic acid to prevent anemia and various fetal developmental disorders in young women (World Health Organization 2018) [2]. The use of iron supplements in combination with folates should be considered in terms of its effectiveness and safety. In our previous experimental study, we found that moderate and long-term supplementation of folic acid affects the level of iron in female micronutrient-deficient rats [3]. Furthermore, we also found that iron and folic acid deficiency, and subsequent supplementation with these micronutrients, affected the process of transcription of folic acid and iron transporters in the duodenum and the liver; however, it did not affect the process of translation of transporter mRNAs to proteins [4,5]. This observation with regard to the iron–folate combination may be beneficial for safety reasons, especially with respect to long-term supplementation because it may inhibit iron overload in the liver and reduce its harmful effects [6].

The effectiveness of iron and folate supplementation depends on many nutritional and environmental factors that affect the bioavailability of vitamins and minerals [7,8]. Among iron absorption inhibitors are phytates, calcium, zinc, and magnesium [9]. The effect of elements (e.g., manganese, vanadium, and zinc) on iron homeostasis may be mediated through absorption, circulation, and regulation of hepcidin production. The majority of the studies have been conducted to show the effect of low or high supplementation of minerals on iron status [10,11]. In addition, there is data to show interactions between iron and minerals; however, there is little information regarding the effect of iron and folic acid supplementation on mineral metabolism. Iron supplementation may sometimes exceed the dietary allowance of iron in women, which may lead to various health problems due to increased bioavailability of free iron. Increased levels of iron may also interfere with the intestinal absorption of bivalent metals.

Therefore, it is important to study the effect of iron overload on the levels of microelements in order to demonstrate the effectiveness and safety of supplementation, which is widely recommended for young women. Therefore, in this study, we aimed to investigate the effect of simultaneous supplementation of iron and folic acid on selected essential and toxic elements in the serum of young women who are deficient in these micronutrients.

## 2. Materials and Methods

### 2.1. Demographic Characteristics

The study protocol was approved by the Ethics Committee at Poznan University of Medical Sciences (approval no. 917/16). The study complies with the ethical standards of the Declaration of Helsinki and its amendments. All subjects provided their written informed consent prior to their inclusion in the study. This study is a part of a project registered at ClinicalTrials.gov no. NCT03438942.

A total of 68 young women were initially screened, and 40 women were finally enrolled for participation. The following were the inclusion criteria: informed written consent, 18–35 years of age, regular menstrual cycles, stable body weight (less than 3 kg self-reported change during the previous 3 months), and diet unchanged during the study. The study was conducted on Polish women (white/Caucasian). It was a homogenous ethnic group. The following were the exclusion criteria: women with chronic diseases and cancer, clinically significant inflammatory processes, abnormal hepatic function, alcohol abuse, drug abuse, smoking, pregnancy, breast-feeding, and use of dietary supplements with minerals in the last 3 months.

### 2.2. Study Design

Participants were divided into two groups: study group (with iron and folate deficiency) and the control group. In all the participants, iron and folic acid status was indicated by the value of unsaturated iron-binding capacity (UIBC) and folic acid level in the blood. Women with UIBC value of above 268 µg/dL and folic acid level of below 7.9 ng/mL were qualified to participate in the study group. The study group received iron and folic acid supplement (14 mg Fe as iron gluconate and 200 µg folic acid per day). The doses of the supplements were selected by the average dose that is present in iron and folic acid supplements available in the Polish market. Moreover, the literature data showed that the dose of daily iron usually used in studies ranged from 10 to 120 mg and the daily dose of folic acid there was between <400 µg to 1000 µg in young women [12,13]. WHO recommends 3 months of iron supplementation also combined with folic acid in menstruating women [12]. The supplement was taken daily at the same time, which was 2 h after the meal for a period of 3 months. During recruitment, the participants underwent medical consultation and examination, and they provided an interview with the gynecologist. Women without any deficiency of iron and folic acid were grouped under the control group, and they did not receive any supplements. All participants were instructed to maintain an isocaloric diet, to continue their previous eating habits, and to perform their routine exercises throughout the conduct of the study. In the study group, blood samples were collected at baseline and after each month, whereas in the control group, blood samples were collected at baseline and after 3 months. Furthermore, 3 days before the collection of blood samples (at the baseline and after 3 months), dietary intake was determined by obtaining 24 h dietary recalls from the subjects. The dietary recall used in the study is recommended by the Polish National Food and Nutrition Institute. All patients recorded their diet during the study. At baseline and after completion of the study, women completed a 3-day food intake record (2 weekdays and 1 weekend day), which was subsequently analyzed by a professional dietician. Nutrient content was determined by a dietician using a computer program (Dietetyk 3.0, Alpha-Net Software, Wroclaw, Poland). All participants completed the study.

### 2.3. Anthropometry

Anthropometric measurements were performed in the morning, 10–12 h after consuming the last meal. For providing measurements, the participants wore light clothes and did not wear shoes. Weight was measured to the nearest 0.1 kg and height was measured to the nearest 1 cm. The body mass index (BMI) was calculated by dividing the weight (kg) by the height (m^2^).

### 2.4. Blood Sampling

Blood samples were collected in the morning after resting in the supine position for 30 min, following a 12-h fast and a night’s rest. Blood was withdrawn from a forearm vein and collected in serum separator tubes to obtain serum, and another sample was collected in tubes containing anticoagulant to obtain whole blood. The coagulated blood was left to clot at room temperature and then centrifuged. The serum was separated and frozen and stored at −80 °C until analysis.

### 2.5. Biochemical Parameters

Morphological parameters were analyzed in the certified diagnostic laboratory. Hemoglobin concentration was measured using the cyanmethemoglobin method (Merck, Germany). Complete blood count was obtained using a hematology analyzer (cel-Dyn 3700, Abbott Laboratories, Lake Bluff, IL, USA). UIBC was analyzed in serum using the photometric reagent ferene (DiaSys, Holzheim Germany) on a Konelab 20i biochemical analyzer (Thermo Scientific, Finland). The level of folate was determined using the electrochemiluminescence method with a Cobas 6000 (Roche Diagnostics, Indianapolis, IN, USA) and reagent Folate III test (Roche Diagnostics GmbH, Mannheim, Germany). The precision and accuracy of the techniques used to assay the biochemical parameters were validated. Reproducibility was checked with a human serum control (Randox Laboratories, Crumlin, UK). Serum hepcidin and ferritin concentrations were determined using a Human ELISA Kit (Wuhan Fine Biotech, Wuhan, China). Plasma homocysteine (Hcy) levels were determined using enzyme-cycling Hcy assay with commercial kits (Diazyme Homocysteine Assay, Diazyme Laboratories, Poway, CA, USA) and a fully automated Konelab 20i Analyzer (Thermo Scientific, Vantaa, Finland). Accuracy was assessed by using the recovery value, which ranged between 95 and 109%. The variability coefficient did not exceed 10%.

### 2.6. Measurement of Elements

Measurement of essential and toxic elements in serum was conducted based on a previous publication [11]. Samples were mineralized in a high-pressure closed microwave digestion system (Ethos One, Milestone, Sorisole, Italy). An Elan DRC II ICP-MS (PerkinElmer SCIEX, Markham, ON, Canada) was used to determine the following elements: Fe, Li, Mg, Ca, Ti, V, Cu, Zn, As, Rb, Sb, Tl, Se, Co, and Sr. Calibration based on a weighted least squares calibration curve was employed for all elements. The linearity—calculated as R^2^—was acceptable for all the analyzed elements (R^2^ > 0.999). The trueness of the analytical method was assessed by analyzing the certified reference material (CRM) Seronorm™Trace Elements Serum L-2 (LGC, Wesel, Germany). The values of recovery were within an acceptable range for all the analytes, which demonstrates that the described analytical procedure is fit for the intended purpose.

### 2.7. Statistical Analysis

The data were analyzed using Statistica 13.0 software (Statsoft, Krakow, Poland). The results are shown as arithmetic mean ± standard deviation (SD) and median. The Shapiro–Wilk test was used to check the normal distribution of the data. A comparison between the differences was made using Friedman’s analysis of variance (ANOVA) and Kendall’s coefficient of concordance. The Wilcoxon rank-sum test was performed to determine the statistical significance of the variables in both groups. To compare the groups, we used the Mann–Whitney test. Associations between variables were calculated based on the Spearman coefficient of correlation. *p* values less than 0.05 were regarded as significant. According to our calculation, a sample size of at least 12 subjects in each group would yield at least 80% power to detect an intervention effect that was statistically significant at the 0.05 α level.

## 3. Results

Table 1 shows the demographic data including data on biochemical and dietary intake of both study and control groups. The age of the participants and their BMI values were comparable in both groups. The supply of macro and micronutrients was similar for both the groups, and the nutritional parameters did not change during the study. According to the results, there was a significant difference between the groups with respect to the level of iron and folate. At baseline, UIBC in the study group was significantly higher than that in the control group. The concentration of ferritin and folic acid in serum were markedly lower in the study group than that of the control group. It was observed that serum levels of ferritin and folate significantly increased after 3 months of supplementation, and the value of these parameters was similar to the control group. The following elements were analyzed in this study: Fe, Li, Ca, Mg Ti, Co, Cu, Zn, Rb, Sb, Sr, Tl, Se, V, and as (Table 2 and Table 3). According to the data, at baseline, the level of Mg and Co were markedly higher in the study group than that of the control group, whereas after the intervention, the level of zinc was significantly higher in the control group than that of the study group (Table 3).

In the study group, the level of iron gradually increased after the supplementation (Figure 1). The level of zinc decreased after the first month of the intervention and its level was maintained until the end of the study period (Figure 2). After 2 months of supplementation, there was a decrease in the level of magnesium and calcium in the study group (Figure 3 and Figure 4).

In the control group, there was a significant decrease in the level of iron after 3 months of the study period (Figure 5). We also analyzed the correlation between the level of iron and other elements. We obtained a significant and inverse correlation between iron and zinc levels in the study group (R = −0.29). It is noteworthy that in the control group the correlation between Fe and Zn was positive (R = 0.29) but not significant (Figure 6).

Vanadium and arsenic increased in both the treatment and control group (Figure 7, Figure 8, Figure 9 and Figure 10) and there was a positive correlation between them (Figure 11).

During this study, the levels of other elements did not change in both groups (Table 2 and Table 3).

## 4. Discussion

Despite the fact that supplementation of the diet with iron and folic acid is recommended for women of childbearing age, little is known about the effect of simultaneous intake of these micronutrients on the bioavailability of other elements. To the best of our knowledge, this is the first study to analyze the effect of iron and folic acid intake on the levels of essential and toxic elements in young women. According to our results, iron and folic acid supplementation disturbed the bioavailability of magnesium, calcium, and zinc in these women, which might prove to be harmful to the developing fetus. Moreover, increased levels of serum iron did not markedly affect the hemoglobin level or the UIBC after 3 months of supplementation in the study group. This might be due to the rapid increase in the levels of serum vanadium and arsenic after the 3 months of supplementation; however, these changes were observed in both groups.

In the study, women of reproductive age between 18 and 35 years were included. Such a life period in European women is associated with rather a stable hormone level and low risk for metabolic disorders [14]. In this population, iron deficiency and also folic acid and Se deficit are frequently observed [15]. In women, age-related changes in trace elements status was shown [16,17]. Generally, in women hair, Hg increased and V decreased with age [16]. In a previous study we found lower hair minerals concentration in women aged 30–39 compared to women aged 19–30 and 41–50 years [17].

According to our previous findings, simultaneous supplementation of iron and folic acid in rats deficient in these micronutrients caused beneficial changes with respect to the level of folates; however, the changes in iron levels were less significant [4,5]. This may be because dietary deficiency of folic acid and iron, and subsequent supplementation with these nutrients affected the process of transcription but not the translation of folic acid and iron transporters in the duodenum and liver [4,5]. Moreover, folic acid levels might affect the level of iron during moderate and long-term supplementation with these micronutrients [6]. Although we did not find any significant changes in UIBC, hemoglobin, or hepcidin level after the intervention, we observed a marked increase in the level of iron, ferritin, and folate in the study group. This shows the beneficial effect of supplementation. However, according to our results, the level of zinc, magnesium, and calcium decreased with the increase in the level of iron in the study group. These changes show the potential interactions between the abruption of different minerals. This interaction might have affected the level of iron and its supplementation in the iron-deficient state. Previous studies have demonstrated the interaction between iron and zinc. This interaction is seen during transportation for distribution, especially with regard to hepcidin activity [10]. The negative effect of iron supplementation on the level of zinc has been recorded in human studies, which shows that excessive levels of iron may decrease the uptake of zinc in nonpregnant and pregnant women taking iron supplements [10,18]. The negative correlation between the level of iron and zinc in the study group confirmed these explanations. Some studies have suggested that folic acid (particularly in high doses) may also reduce zinc absorption and impair its utilization [19]. Shankar et al. [20] found that weekly supplementation of iron and folic acid developed hypozincemia both in anemic and nonanemic pregnant women. In this study, we obtained similar results—magnesium levels decreased after a daily supplementation of folic acid and iron.

Literature also describes the interaction between magnesium, calcium, and iron, which is mainly concerned with the absorption and ion shift between tissues and redistribution in an organism [21,22,23]. There is more information regarding the effect of elements such as calcium and magnesium on the absorption of iron absorption. For example, a study shows that calcium affects the uptake of iron through divalent metal transporter 1 (DMT1) in the intestinal lumen and also showed that it may inhibit the transfer of iron through exporter ferroprotein (FPN) and hephestin [22].

The effect of calcium on iron absorption and transfer depended on Ca:Fe ratio and impact time, because the adaptation process was shown with time [22]. Another in vitro study showed that calcium and zinc alter iron metabolism by increasing the activity of FPN, which caused a reduced net iron absorption [11]. The effect of iron overload on calcium malabsorption has been observed in thalassemia [24]. It was found that iron hyperabsorption leads to impaired calcium transport by deregulation of calciotropic hormone production and response, low transcellular calcium uptake, aberrant hepcidin release and response, and overexpression of DMT1 and/or FPN-1. This negative interaction may also include negative regulators of calcium absorption, such as FGF-23 [24].

In a similar study conducted by Tiwari et al. [25], hemoglobin level was found to be significantly increased in pregnant women with anemia after supplementation with iron and folic acid. However, in the aforementioned study, the level of hemoglobin at baseline was lower than that in this study (their inclusion criteria was hemoglobin level of <11.0 g/dL). In addition, the daily dose of iron and folic acid was much higher (100 mg and 500 µg, respectively) in the aforementioned study than that of ours. Tiwari et al. [25] also reported significantly increased levels of copper and slightly increased levels of zinc after the intervention. They also found that iron deficiency anemia leads to a decrease in the level of essential trace minerals and that iron and folic acid supplementation recovered their levels. Our findings did not confirm the results obtained for copper, zinc, and selenium. The observed noncompliance might be the result of other criteria for inclusion in the studies.

Despite the observed interaction, a multimineral tablet containing iron and enriched with zinc and folate is usually recommended in reducing anemia [26]. This issue has been discussed in our previous study [6], where we found that folic acid decreases the efficiency of iron supplementation and at the same time, we concluded that folic acid in the diet inhibited iron overload in the liver and reduced Fe-catalyzed oxidative stress. Another study indicates that zinc supplemented with iron activates metallothionein and scavenging of hydroxyl radicals, which helps to control oxidative stress caused by excess iron [18]. A double-blinded clinical trial (JiVitA-3) conducted on a large population of pregnant women showed a better effect of supplementation with multiple micronutrients than that of iron and folic acid alone in improving the status of micronutrients in pregnant women [27].

It is noteworthy that the dietary intake of calcium, magnesium, iron, zinc, and folic acid in both groups was below what was recommended. These results are consistent with our other studies with the participation of young women [17,28]. At baseline, we found that the average concentration of magnesium and calcium was slightly lower than that of the reference value. In the study group, the level of zinc was within the normal range before the intervention, but during supplementation, it decreased to the lower limit of the normal range. This shows that the dietary intake and the status of minerals in young Polish women were rather low. It should be emphasized that low dietary supply of macro and microelements and low status of these nutrients may intensify the effect of the interactions [21,29]. Probably, low content of iron in the diet for a long time without supplementation may lead to its deficiency in young women.

In this study, no toxic levels were found for any of the analyzed elements. However, we observed significantly increased levels of arsenic and vanadium in the third month of supplementation in both groups. This result suggests that environmental factors along with the supplementation might have an effect on the study population. All examined women lived in the area of Poznan and the surrounding area and they may be frequently exposed to environmental pollution. Moreover, all the study participants were tested during the same time of the year; it was during the turn of autumn and winter of 2016 and 2017. During this period, due to low temperatures, every household used heating systems, which may depend on coal, gas, or electricity. Air contains a mixture of arsenite and arsenate, with a negligible amount of organic arsenic species [3]. A previous study has shown the relationship between high levels of arsenic and the living areas where coal was used as fossil fuels [30,31]. Vanadium is present in both crude oil and coal, and a previous study has demonstrated its harmful effects on the human body [32,33]. They showed a positive correlation between vanadium and arsenic content in the whole population, which confirmed that exposure to these elements came from an external source. Element levels in serum and urinary were related also to sampling seasons in pregnant women [34]. In the mentioned study, folic acid and iron supplementation influenced urinary Cs, Mo, and Sb concentration without an effect on As and V. Vanadium and arsenic show chemical similarities and both may be toxic to humans. Diet and drinking water are the main sources of exposure to As and V for the general population [35]. The chemical structure and content of As and V in soil, water, and food may be associated with their interaction on the absorption level and the distribution level in the organism, and this may result in a positive correlation in the blood of young women.

This study has some limitations. First, the study group was small, which might be due to the rigorous inclusion and exclusion criteria and also due to the staged studies that required high commitment and compliance with time and nutrition regimes. In this study, we involved data of only those women who participated till the end of the study period and who completed all stages of the study. Second, we assessed the levels of selected essential and toxic elements in serum. The results of arsenic and vanadium levels surprised us because we had collected hair and nails samples from women, and these samples have often been used as indicators of exposure to inorganic elements. Moreover, there is no placebo group involving women with folic acid and iron deficiency because of ethical reasons. Taking placebo by women with a deficiency may be harmful to their health and in the future may have a negative influence on their offspring.

## 5. Conclusions

The simultaneous supplementation of iron and folic acid for 3 months improved their status and is more beneficial in terms of folic acid than that of iron in the women of childbearing age. Furthermore, the simultaneous supplementation of iron and folic acid impairs the level of zinc, calcium, and magnesium in these women. Therefore, based on these results, we conclude that recommendations for the preventive and therapeutic use of folic acid and iron should consider the adequate supply of micro and macroelements from the diet and supplements. Moreover, environmental factors should also be considered when assessing the levels of bioelements, especially toxic bioelements.

## Figures and Tables

**Figure 1 ijerph-18-01360-f001:**
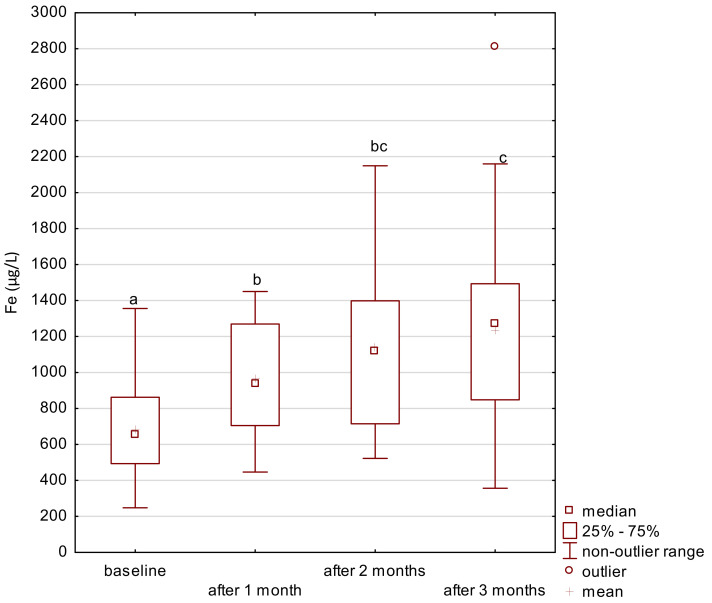
Iron concentration in serum in the study group. a,b,c significantly different; Friedman’s ANOVA, *p* = 0.003.

**Figure 2 ijerph-18-01360-f002:**
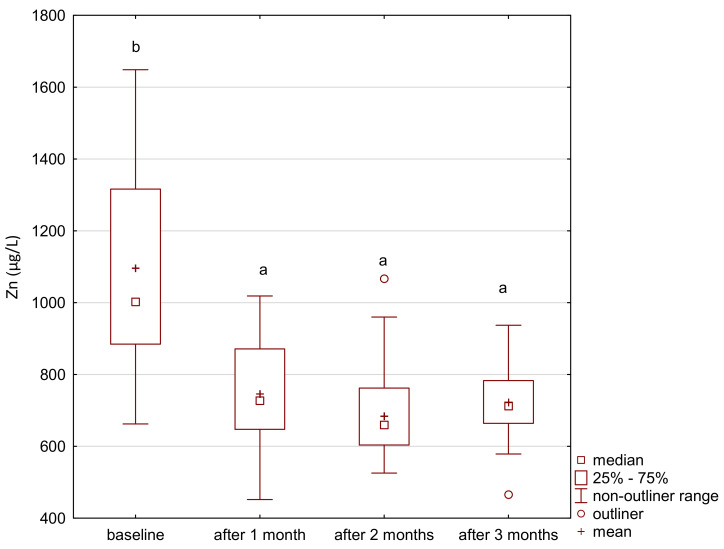
Zinc concentration in serum in the study group. a,b significantly different; Friedman’s ANOVA, *p* < 0.001.

**Figure 3 ijerph-18-01360-f003:**
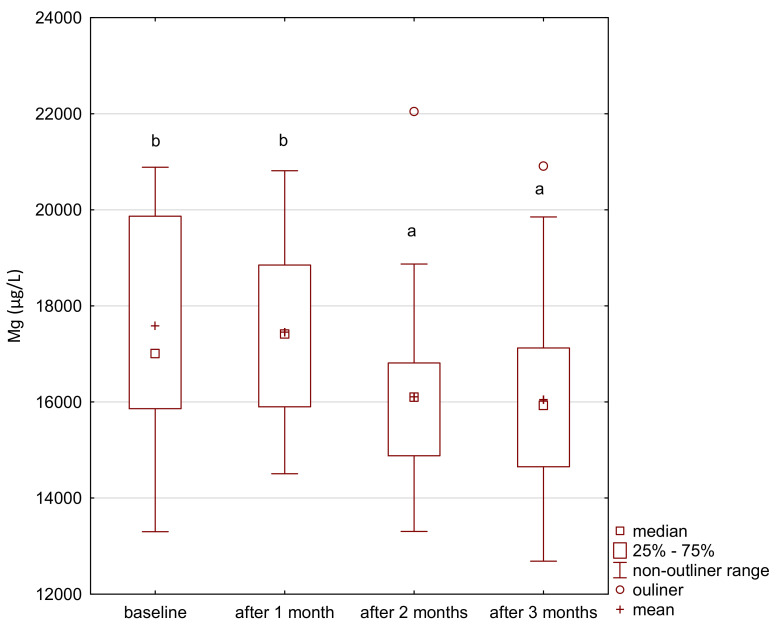
Magnesium concentration in serum in the study group. a,b significantly different; Friedman’s ANOVA, *p* = 0.014.

**Figure 4 ijerph-18-01360-f004:**
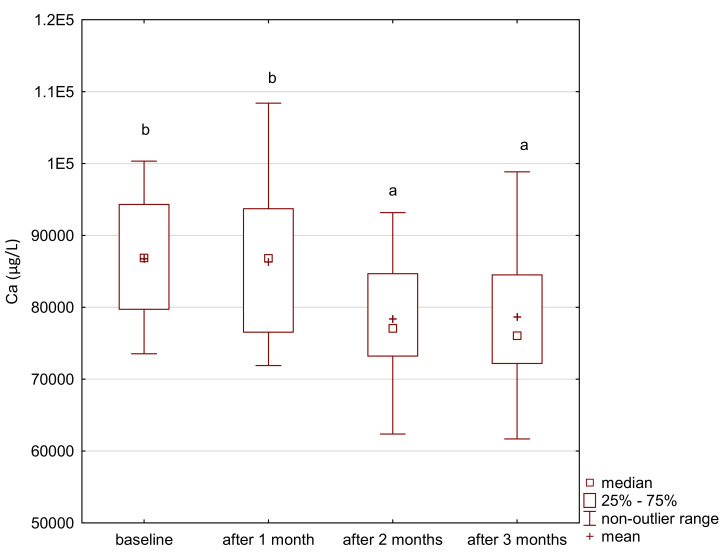
Calcium concentration in serum in the study group. a,b significantly different.

**Figure 5 ijerph-18-01360-f005:**
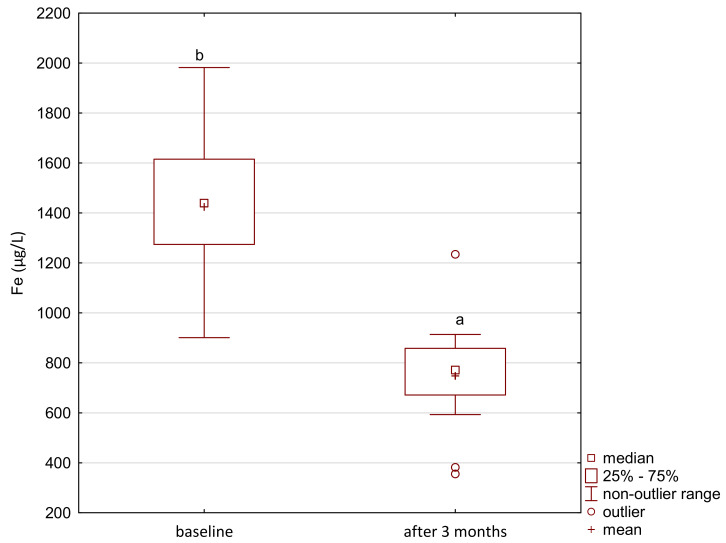
Iron concentration in serum in the control group; Wilcoxon rang-sum test, *p* = 0.0003; #3 baseline significantly different with the study group, Mann–Whitneytest *p* < 0.001; *3 after 3 months significantly different with the study group, Mann–Whitney test *p* = 0.001.

**Figure 6 ijerph-18-01360-f006:**
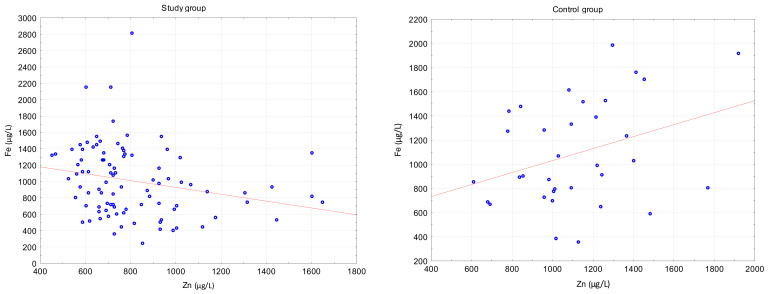
Spearman correlation between iron and zinc serum concentration in study group (R = −0.29, *p* = 0.03) and control group (R = 0.29; *p* > 0.05).

**Figure 7 ijerph-18-01360-f007:**
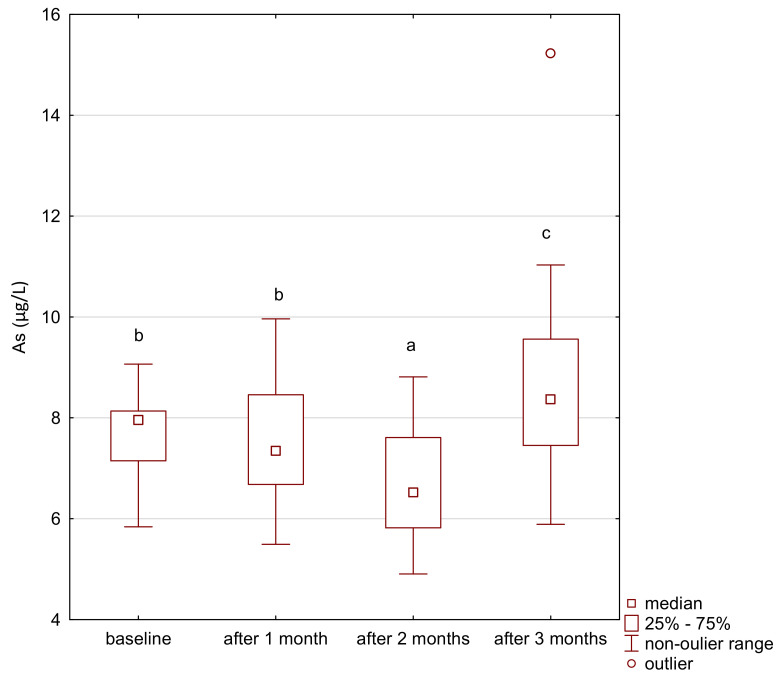
Arsenic concentration in serum in the study group. a,b,c significantly different;. Friedman’s ANOVA, *p* = 0.007.

**Figure 8 ijerph-18-01360-f008:**
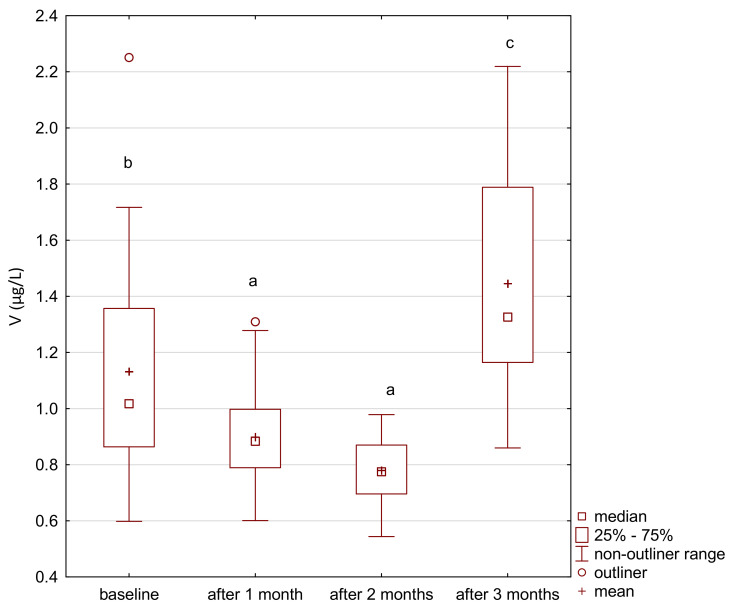
Vanadium concentration in serum in the study group. a,b significantly different; Friedman’s ANOVA, *p* < 0.001.

**Figure 9 ijerph-18-01360-f009:**
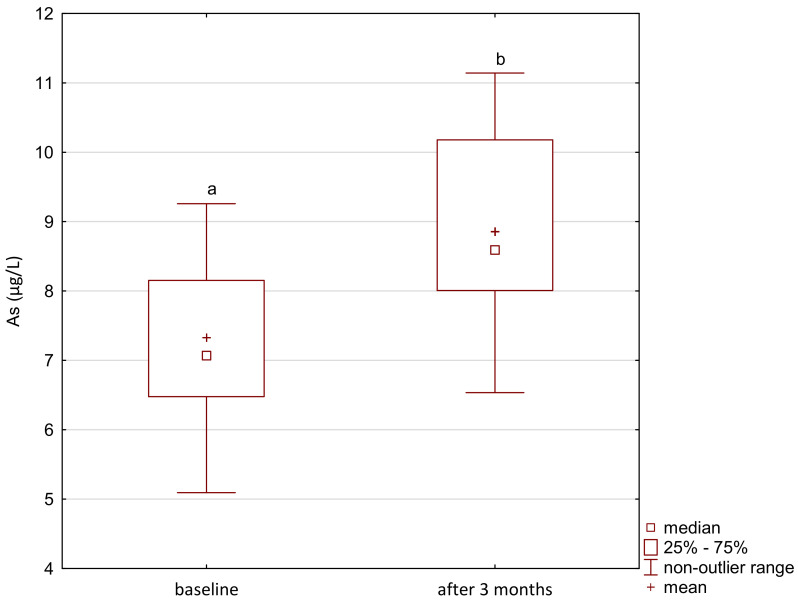
Arsenium concentration in serum in the control group. a,b significantly different Wilcoxon rang-sum test, *p* = 0.006.

**Figure 10 ijerph-18-01360-f010:**
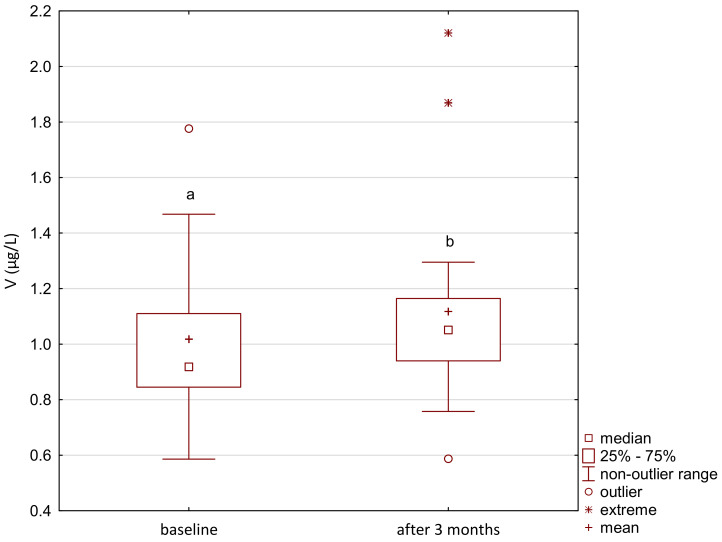
Vanadium concentration in serum in the control group. a,b significantly different; Wilcoxon rang-sum test, *p* = 0.04.

**Figure 11 ijerph-18-01360-f011:**
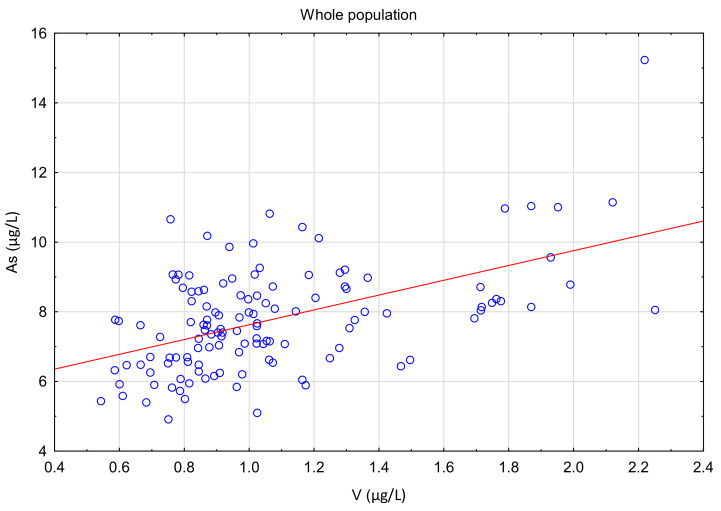
Spearman correlation (R = 0.46; *p* < 0.01) between arsenic and vanadium serum level in whole population.

**Table 1 ijerph-18-01360-t001:** Age, BMI and biochemical and dietary parameters in the study and control groups; mean ± SD/median.

	Study Group (n = 23)		*p*-Value	Control Group (n = 17)		*p*-Value
	Baseline	After 3 Months		Baseline	After 3 Months	
Age (years)	25.2 ± 3.5 25			26.1 ± 3.6 26		
BMI (kg/m^2^)	20.6 ± 1.9 20.5	20.5 ± 1.9 20.3	NS	21.9 ± 2.0 21.3	21.7 ± 2.0 21.2	NS
Rbc (×10^6^/µL)	4.35 ± 0.25 4.42	4.39 ± 0.26 4.46	NS	4.37 ± 0.25 4.42	4.36 ± 0.31 4.38	NS
Hb (g/dL)	12.93 ± 0.84 13.10	13.19 ± 0.74 13.3	NS	13.3 ± 0.47 13.20	13.09 ± 0.67 13.2	NS
Hct (%)	37.87 ± 2.09 38.50	38.44 ± 1.83 38.50	NS	38.31 ± 1.36 38.40	37.96 ± 1.78 38.20	NS
UIBC (µg/dL)	272.24 ± 124.37 278.13 ^#1^	277.49 ± 84.88 260.61	NS	205.22 ± 56.98 210.95 ^#1^	253.28 ± 65.72 256.31	NS
Hepcidin (µg/L)	29.07 ± 8.44 27.97	25.43 ± 5.33 23.41	NS	28.27 ± 8.31 28.70	25.03 ± 4.48 25.52	NS
Ferritin (µg/L)	22.63 ± 13.08 19.27 ^a,#2^	38.35 ± 16.21 36.47 ^b^	0.003	37.51 ± 19.34 36.19 ^#2^	37.76 ± 17.57 36.97	NS
Homocistein (µmol/L)	9.51 ± 2.69 9.32	9.07 ± 1.97 8.83	NS	8.70 ± 2.94 8.01	9.46 ± 3.02 9.10	NS
Folate (ng/mL)	5.82 ± 1.64 5.98 ^a,#3^	10.23 ± 5.41 10.83 ^b^	0.001	9.81 ± 2.98 10.51 ^#3^	8.63 ± 5.12 9.39	NS
Dietary intakes						
Energy (kcal)	1703 ± 431 1679	1486 ± 274 1458	NS	1765 ± 467 1765	1624 ± 374 1555	NS
Protein (% energy)	16.35 ± 5.35 15.20	18.22 ± 5.75 16.48	NS	16.64 ± 3.40 16.11	18.58 ± 3.21 18.60	NS
Fat (% energy)	35.09 ± 6.54 36.00	33.78 ± 6.78 35.55	NS	33.92 ± 7.67 36.30	31.26 ± 6.86 32.04	NS
Carbohydrate (% energy)	55.12 ± 32.34 48.67	49.84 ± 10.67 48.00	NS	49.45 ± 7.97 48.46	47.95 ± 7.67 48.44	NS
Fibre (g)	23.24 ± 16.61 20.41	19.54 ± 7.42 20.86	NS	27.46 ± 13.99 23.90	26.80 ± 17.94 23.66	NS
Calcium (mg)	580.48 ± 227.85 510.97	642.83 ± 184.37 615.05	NS	606.83 ± 269.23 543.46	668.49 ± 366.86 611.47	NS
Magnesium (mg)	273.49 ± 36.59 279.77	273.40 ± 70.89 272.47	NS	421.36 ± 370.51 334.40	324.20 ± 100.65 329.24	NS
Iron (mg)	10.56 ± 2.35 10.50	10.07 ± 2.74 9.21	NS	11.92 ± 3.80 10.78	13.74 ± 8.43 11.52	NS
Zinc (mg)	8.53 ± 2.34 8.49	8.32 ± 1.82 7.63	NS	9.76 ± 3.14 9.49	10.91 ± 5.26 10.30	NS
Copper (mg)	1.14 ± 0.22 1.09	1.11 ± 0.30 1.05	NS	1.47 ± 0.62 1.23	1.37 ± 0.59 1.31	NS
Folate (µg)	309.94 ± 151.96 286.95	305.81 ± 138.63 313.52	NS	315.66 ± 147.79 302.83	420.23 ± 270.18 375.32	NS

^a,b^ sugnificantly different Wilcoxon rank-sum test; ^#^ significantly different, Mann–Whitney test, ^#1^
*p* = 0.002; ^#2^
*p* = 0.001; ^#3^
*p* = 0.001. NS, non significant.

**Table 2 ijerph-18-01360-t002:** Elements concentration in the study group (without significantly changes); mean ± SD/median.

Element (µg/L)	Baseline	After 1 Month	After 2 Months	After 3 Months	*p*-Value
Li	1.29 ± 1.15 0.81	0.94 ± 0.51 0.92	1.03 ± 0.44 0.97	0.88 ± 0.46 0.77	NS
Ti	125.50 ± 76.07 71.72	68.54 ± 12.24 66.98	73.81 ± 19.30 69.76	61.04 ± 8.81 61.24	NS
Co	0.67 ± 0.37 0.59	0.55 ± 0.16 0.50	0.45 ± 0.12 0.44	0.51 ± 0.23 0.47	NS
Cu	1065.63 ± 324.49 981.72	1029.73 ± 362.10 922.44	991.34 ± 309.47 931.01	1050.98 ± 370.61 953.33	NS
Rb	180.80 ± 54.87 166.62	156.12 ± 26.09 152.67	157.34 ± 47.48 137.76	170.03 ± 39.60 162.71	NS
Sb	1.93 ± 1.09 1.47	1.46 ± 0.43 1.39	1.96 ± 0.26 2.06	1.41 ± 0.35 1.38	NS
Sr	54.35 ± 30.87 40.60	28.34 ± 9.46 26.55	26.45 ± 7.26 25.51	30.17 ± 7.26 28.97	NS
Tl	0.03 ± 0.01 0.01	0.01 ± 0.01 0.02	0.02 ± 0.01 0.02	0.02 ± 0.01 0.03	NS
Se	96.59 ± 16.86 95.25	91.80 ± 14.60 89.06	87.80 ± 17.28 85.40	89.50 ± 20.47 86.32	NS

NS, non-significant.

**Table 3 ijerph-18-01360-t003:** Elements concentration in the control group; mean ± SD/median.

Element (µg/L)	Baseline	After 3 Months	*p*-Value
Li	1.01 ± 0.70 0.80	1.53 ± 1.93 0.70	NS
Mg	16,035 ± 2168 15,264 ^#1^	15,849 ± 2573 15,109	NS
Ca	82,370 ± 10,501 82,054	77,079 ± 9549 76,179	NS
Ti	110.45 ± 68.74 70.24	61.61 ± 9.39 61.34	NS
Co	0.44 ± 0.13 0.39 ^#2^	0.41 ± 0.11 0.36	NS
Cu	1157.84 ± 339.79 1093.67	1156.39 ± 376.21 1032.74	NS
Zn	1161.73 ± 293.12 1152.93	1065.33 ± 295.17 1008.26 ^*1^	NS
Rb	203.96 ± 58.44 181.85	182.06 ± 46.64 162.47	NS
Sb	2.95 ± 1.73 2.87	1.42 ± 0.40 1.46	NS
Sr	45.39 ± 23.36 37.86	33.06 ± 11.39 28.68	NS
Tl	0.04 ± 0.02 0.03	0.02 ± 0.01 0.02	NS
Se	104.79 ± 20.00 107.64	95.61 ± 19.33 90.41	NS

NS, non-significant; ^#^ significantly different with the study group, Mann–Whitney test; ^#1^
*p* = 0.04; ^#2^
*p* = 0.004; * significantly different with the study group, Mann–Whitney test; ^*1^
*p* = 0.0001.

## Data Availability

The data supporting the findings of this study are available on reasonable request from the corresponding author.

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
