# Peer review of "Effect of Iron and Folic Acid Supplementation on the Level of Essential and Toxic Elements in Young Women"

_ijerph, 2021, doi:10.3390/ijerph18031360_

Round 1
Reviewer 1 Report
The study entitled “Effect of iron and folic acid supplementation on the level of essential and toxic elements in young women” is aimed to aimed to investigate the effect of iron and folic acid supplementation on the levels of selected essential and toxic elements in the serum of micronutrient-deficient young women. The manuscript presents many limitations as evidenced by the authors; not least observations and details about micronutrients deficency levels identification. Anyway the study if extensively revised can contribute in the understanding of potential interactions between micronutrients supplementation such as iron and folic acid in women of childbearing age and the interactions with toxic elements.
- Line 85: Authors reported that: “14 mg Fe as iron gluconate and 200 µg folic acid per day”. Are the doses and the duration of exposure selected by guideline or supported by literature? Could Authors add informations about
- From line 82: Could Authors add details about deficency micronutrients levels identification?
- Could Authors provide informations about the women’ origin ethnic group?and if not
homogeneous provide a discussion.
- The authors should discuss in light of literature the wide women’s age range selected in relation to different hormonal and metabolic status also respect to analyzed elements
- Authors are invited to present all the data set for Iron, Arsemium, Vanadium showing each experimental time.
- I suggest to join Results and Discussion sections to improve text fluency and to give a more informative view of the overall results
Minor Comment:
figures and figure captions are not right formatted.
Line 50; 265; 242; 298 and 317 Please, Check references in brackets
Author Response
Dear Editor and Reviewers,
We are very grateful for your comments on our manuscript. We have revised the manuscript in accordance with your advice.
Reviewer 1
- Line 85: Authors reported that: “14 mg Fe as iron gluconate and 200 µg folic acid per day”. Are the doses and the duration of exposure selected by guideline or supported by literature? Could Authors add informations about.
Response: The doses of the supplements are dictated by the average dose that is present in iron and folic acid supplements available in the Polish market. This information has been added to the text.
- From line 82: Could Authors add details about deficency micronutrients levels identification?
Response: These details are in the sentences: „In all the participants, iron and folic acid status was indicated by the value of unsaturated iron-binding capacity (UIBC) and folic acid level in the blood. Women with UIBC value of above 268 µg/dL and folic acid level of below 7.9 ng/mL were qualified to participate in the study group.”
- Could Authors provide informations about the women’ origin ethnic group? and if not
homogeneous provide a discussion.
Response: The study was conducted on Polish women (white/Caucasian). It was a homogenous ethnic group. This information has been added to the text.
- The authors should discuss in light of literature the wide women’s age range selected in relation to different hormonal and metabolic status also respect to analyzed elements.
Response: This problem has been discussed and the text has been added in part “discussion”:
“In the study women of reproductive age between 18 and 35 years were included. Such a life period in European women is associated with rather a stable hormone level and low risk for metabolic disorders [12]. In this population iron deficiency, and also folic acid and Se deficit are frequently observed [13]. In women, it was shown age-related changes in trace elements status [14][15]. Generally, in women hair, Hg increased and V decreased with age [14]. In a previous study we found lower hair minerals concentration in 30-39 age women compared to women aged 19-30 and 41-50 years [15].”
- Authors are invited to present all the data set for Iron, Arsemium, Vanadium showing each experimental time.
Response: In the control group there were two stages: at baseline and after three months. This information was included in ‘the study design’: “In the study group, blood samples were collected at baseline and after each month, whereas in the control group, blood samples were collected at baseline and after 3 months.”
- I suggest to join Results and Discussion sections to improve text fluency and to give a more informative view of the overall results
Response: The information for authors includes information about the main sections in Research manuscripts as follows: Introduction, Materials and Methods, Results, Discussion, Conclusions. According to the guidelines for authors, we decided not to combine Results and Discussion.
- figures and figure captions are not right formatted.
Response: Figures and captions were formatted according to the instruction for authors:
- File for Figures and Schemes must be provided during submission in a single zip archive and at a sufficiently high resolution (minimum 1000 pixels width/height, or a resolution of 300 dpi or higher). Common formats are accepted, however, TIFF, JPEG, EPS and PDF are preferred.
- All Figures, Schemes and Tables should be inserted into the main text close to their first citation and must be numbered following their number of appearance (Figure 1, Scheme I, Figure 2, Scheme II, Table 1, etc.).
- All Figures, Schemes and Tables should have a short explanatory title and caption.
The size of the figures has been reduced.
- Line 50; 265; 242; 298 and 317 Please, Check references in brackets
Response: All references have been checked.
Reviewer 2 Report
The manuscript presents a significant study with very interesting results for the scientific community specially clinal practicians. The authors have also been very carefull describing the limitations of the study but looking at their previous research they will keep on researching on this topic increasing the knowledge.
Methodology is correct and the design of the study is fine.
No reference to the country and/or city/region of the study has been made so a short sentence about this should be inserted.
Considering the ICP method, the authors could have justifies why they chose those elements and not others that the ICP-OES surely determines at the same time like Pb and Cd.
Figures are of great quality and relevance although their sizes could be finally reduced.
The discussion about the positive correlation between As and V is quite innovative and interesting although this mechanism of action/interaction/toxicity could have been better discussed and not only referred to environmental origin which might be difficult to correlate.
Line 328: "they health" should be changed for "their health".
Author Response
Dear Editor and Reviewers,
We are very grateful for your comments on our manuscript. We have revised the manuscript in accordance with your advice.
Reviewer 2
- No reference to the country and/or city/region of the study has been made so a short sentence about this should be inserted.
Response: In Materials and Methods we have added the sentence: “The study was conducted on Polish women (white/Caucasian). It was a homogenous ethnic group.”
- Considering the ICP method, the authors could have justifies why they chose those elements and not others that the ICP-OES surely determines at the same time like Pb and Cd.
Response: In the blood samples we measured also: Be, Al, Mn, Cr, Ni, Ga, Mo, Ag, Cd, Cs, Ba, Pb, U, however the concentration of these elements in the most of samples were below the limit of detection (<LOD). In this study, we selected those elements with the detectable level.
- Figures are of great quality and relevance although their sizes could be finally reduced.
Response: The size of the figures has been reduced.
- The discussion about the positive correlation between As and V is quite innovative and interesting although this mechanism of action/interaction/toxicity could have been better discussed and not only referred to environmental origin which might be difficult to correlate.
Response: The positive correlation between As and V has been discussed broader:
“Element levels in serum and urinary were related also to sampling seasons in pregnant women [32]. In the mentioned study folic acid and iron supplementation influenced urinary Cs, Mo, and Sb concentration without effect on As and V. Vanadium and arsenic show chemical similarities and they both may be toxic to humans. Diet and drinking water are the main sources of exposure to As and V for the general population [33]. The chemical structure and content of As and V in soil, water, and food may be associated with their interaction on the absorption level and the distribution level in the organism and this may result in a positive correlation in the blood of young women.”
- Line 328: "they health" should be changed for "their health".
Response: It has been changed.
Reviewer 3 Report
I found this to be a well designed and performed and well described study. I have only one comment. The number of significant figures used in the tables are often grossly too large. For example, the calcium concentrations could not possibly have been determined to five significant figures. The number of significant numbers must actually and accurately represent the precision of each experimental measurement.
Author Response
Dear Editor and Reviewers,
We are very grateful for your comments on our manuscript. We have revised the manuscript in accordance with your advice.
Reviewer 3
- The number of significant figures used in the tables are often grossly too large. For example, the calcium concentrations could not possibly have been determined to five significant figures. The number of significant numbers must actually and accurately represent the precision of each experimental measurement.
Response: The number of significant figures for age, BMI, energy intake, serum Ca and Mg concentrations have been changed in the tables.
Round 2
Reviewer 1 Report
The authors answer to all comments and improved the information in the text.
Only the answer to my first request continues to be only partially addressed. I try to ask the question differently:
Could the authors to comment how they chose 3 months as experimental time? Is it reported in the literature to be a sufficient (minimum) time to compensate iron and folate deficiency or other reasons? Moreover, beyond what is available in Polish market, can the authors introduce guide-line or literature-based commentary on iron and folate supplement doses?
Author Response
Dear Editor and Reviewer,
We are very grateful for your comments on our manuscript. We have revised the manuscript in accordance with your advice.
Reviewer 1
- Could the authors to comment how they chose 3 months as experimental time? Is it reported in the literature to be a sufficient (minimum) time to compensate iron and folate deficiency or other reasons? Moreover, beyond what is available in Polish market, can the authors introduce guide-line or literature-based commentary on iron and folate supplement doses?
Response: We have added the following sentences to the Materials and methods: “Moreover, the literature data showed that the dose of daily iron usually used in studies ranged from 10 to 120 mg and the daily dose of folic acid there was between <400 µg to 1000 µg in young women [13][12] WHO recommends 3 months of iron supplementation also combined with folic acid in menstruating women [12].”